

# Differentiating wild from captive animals: an isotopic approach

Luiza Brasileiro[1,2], Rodrigo Ribeiro Mayrink[2,3], André Costa Pereira[2], Fabio José Viana Costa[4] and Gabriela Bielefeld Nardoto[2]

[1] Diretoria de Fiscalização Ambiental, Brasilia Ambiental, Brasília, DF, Brazil
[2] Departamento de Ecologia, Instituto de Ciências Biológicas, Universidade de Brasília, Brasília, DF, Brazil
[3] Setor Técnico-Científico, Policia Federal, Belo Horizonte, MG, Brazil
[4] Instituto Nacional de Criminalística, Policia Federal, Brasília, DF, Brazil

Corresponding authors
Luiza Brasileiro,
brasileiro.luiza@gmail.com
Gabriela Bielefeld Nardoto,
gbnardoto@unb.br

## ABSTRACT

**Background:** Wildlife farming can be an important but complex tool for conservation. To achieve conservation benefits, wildlife farming should meet a variety of criteria, including traceability conditions to identify the animals' origin. The traditional techniques for discriminating between wild and captive animals may be insufficient to prevent doubts or misdeclaration, especially when labels are not expected or mandatory. There is a pressing need to develop more accurate techniques to discriminate between wild and captive animals and their products. Stable isotope analysis has been used to identify animal provenance, and some studies have successfully demonstrated its potential to differentiate wild from captive animals. In this literature review, we examined an extensive collection of publications to develop an overall picture of the application of stable isotopes to distinguish between wild and captive animals focusing on evaluating the patterns and potential of this tool.

**Survey methodology:** We searched peer-reviewed publications in the Web of Science database and the references list from the main studies on the subject. We selected and analyzed 47 studies that used $\delta^{13}C$, $\delta^{15}N$, $\delta^2H$, $\delta^{18}O$, and $\delta^{34}S$ in tissues from fish, amphibians, reptiles, birds, and mammals. We built a database from the isotope ratios and metadata extracted from the publications.

**Results:** Studies have been using stable isotopes in wild and captive animals worldwide, with a particular concentration in Europe, covering all main vertebrate groups. A total of 80.8% of the studies combined stable isotopes of carbon and nitrogen, and 88.2% used at least one of those elements. Fish is the most studied group, while amphibians are the least. Muscle and inert organic structures were the most analyzed tissues (46.81% and 42.55%). $\delta^{13}C$ and $\delta^{15}N$ standard deviation and range were significantly higher in the wild than in captive animals, suggesting a more variable diet in the first group. $\delta^{13}C$ tended to be higher in wild fishes and in captive mammals, birds, reptiles, and amphibians. $\delta^{15}N$ was higher in the wild terrestrial animals when controlling for diet. Only 5.7% of the studies failed to differentiate wild and captive animals using stable isotopes.

**Conclusions:** This review reveals that SIA can help distinguish between wild and captive in different vertebrate groups, rearing conditions, and methodological designs. Some aspects should be carefully considered to use the methodology properly, such as the wild and captivity conditions, the tissue analyzed, and how

homogeneous the samples are. Despite the increased use of SIA to distinguish wild from captive animals, some gaps remain since some taxonomic groups (*e.g.,* amphibians), countries (*e.g.,* Africa), and isotopes (*e.g.,* $\delta^2$H, $\delta^{18}$O, and $\delta^{34}$S) have been little studied.

## INTRODUCTION

Human activities are the most relevant causes of defaunation, mainly habitat degradation and species overexploitation (*Young et al., 2016*). The maintenance and management of wild species in captivity can be an important tool for conservation, either by maintaining gene banks or target species that would be unlikely to survive in the natural environment (*CDB, 1992*; *Maxted, 2013*) or by decreasing the pressure on wild population by wildlife farming (*Bulte & Damania, 2005*; *Damania & Bulte, 2007*; *Nogueira & Nogueira-Filho, 2011*; *Rizzolo, 2020*).

Wildlife farming is a broad term related to the domestication, production, trade, and consumption of live animals and their products, involving a variety of contexts and species born or raised in captivity (*Rizzolo, 2020*). The legalization of the production and trade of wild animals can be considered a form of supply-side conservation, avoiding natural stock depletion and deforestation, which benefits the recovery and maintenance of the wild population (*Bulte & Damania, 2005*; *Damania & Bulte, 2007*; *Nogueira & Nogueira-Filho, 2011*; *Phelps, Carrasco & Webb, 2014*).

However, wildlife farming is not a simple and uncontentious conservation strategy (*Tensen, 2016*; *Janssen & Chng, 2018*; *Challender et al., 2019*). The increase in the wildlife market brings concerns about production control, management, and trade (*Lyons & Natusch, 2011*; *García-Díaz et al., 2015*; *Janssen & Chng, 2018*). There are a variety of biophysical, market, and regulatory conditions that wildlife farming should meet to achieve conservation benefits (*Phelps, Carrasco & Webb, 2014*; *Tensen, 2016*; *Janssen & Chng, 2018*; *Challender et al., 2019*). In turn, regulatory controls are often based on certification and traceability requirements to identify the animals' origin preventing incorrect or fraudulent statements, alien species invasion, and minimizing damage to wild populations and human health (*European Commission (EC) Notice, 2022*; *Smith et al., 2017*; *Stärk et al., 2019*).

One of the main problems in regulating wildlife farming is identifying the traded animals' real origin (wild or captive) (*Tensen, 2016*; *European Commission (EC) Notice, 2022*). The traditional techniques for discriminating between wild and captive animals, such as trader declarations, government-issued licenses, bands, or microchips, may not be sufficient to prevent doubts or misdeclaration (*Lyons & Natusch, 2011*; *Livingstone & Shepherd, 2016*; *Tensen, 2016*). The efficiency of such methods is even more limited when labels are not expected or mandatory, such as in the trade of animal parts or aquaculture products in most countries, and to identify unmarked escaped or released animals

(*e.g.*, *Dempson & Power, 2004*; *Hammershøj, Asferg & Kristensen, 2004*; *Oceana, 2015*). Thus, there is a pressing need to develop more accurate techniques to discriminate between wild and captive animals and their products.

Stable isotopes are endogenous markers capable of identifying the origin of various samples, such as food, water, or organic materials. The technique has become consolidated as biomarkers of animals' geographic origin (*Hobson & Wassenaar, 2019*). It has also helped track individuals' ecological traits, such as habitat use and diet (*Shipley & Matich, 2020*), making stable isotopes a potential tool for identifying the origin of animals also regarding the rearing system: wild or captive (*Camin et al., 2016*; *Truonghuynh, Li & Jaganathan, 2020*).

Stable isotope analysis (SIA) is based on the variation in the ratio between an element's light and heavy forms (expressed by the letter $\delta$) in response to environmental patterns. As animals access and metabolize environmental resources, their tissues incorporate and express the natural variability in the stable isotopic ratios. Typically, $\delta^{13}C$, $\delta^{15}N$, and $\delta^{34}S$ are categorized as local spatial assays, and their variation in animal tissues reflects their diet composition, such as the primary energy source (*e.g.*, $C_3$ or $C_4$ plants; marine or terrestrial resources) and trophic position, while $\delta^2H$ and $\delta^{18}O$ reflect geographic origin and movement in response to hydrological process (*Hobson & Clark, 1992*; *Fry, 2008*; *Hobson & Wassenaar, 2019*).

However, stable isotope ratios in animal tissues may be influenced by complex physiological processes leading to changes in the isotopic signal compared to the diet (diet-tissue fractionation) and between different tissues (tissue-tissue fractionation). The isotopic fractionation and the speed of incorporation in animals' organic structures, in turn, depends on several environmental, nutritional, and physiological factors, such as tissue turnover rate (*Tieszen et al., 1983*; *Hobson & Clark, 1992*; *Vander Zanden et al., 2015*), diet composition (*DeNiro & Epstein, 1977*; *Magozzi et al., 2019*; *Whiteman et al., 2021*), reproductive and nutritional states (*Doi, Akamatsu & González, 2017*; *Shipley & Matich, 2020*). Comparing wild and captive animals using stable isotopes should consider additional points to avoid confounding effects, such as the characteristics of captivity (*e.g.*, intensive or extensive farming), and whether the animal changed the captive/wild state and the timescales involved.

There are a variety of wildlife farming modalities according to different criteria, such as the management intensity (herding, ranching, and farming) (*Hudson, 1989*) or according to the system used to produce specimens (born in captivity, bred in captivity or ranched) (CITES classification; *Lyons, Natusch & Jenkins, 2017*). These modalities can subject captive animals to significantly different conditions, influencing their isotopic signature. However, the potential of SIA to differentiate wild from captive animals increases under the general assumption that individuals are likely to access different resource items from distinct geographic origins in these two environments (*e.g.*, *Dempson & Power, 2004*; *Kays & Feranec, 2011*; *Chaguri et al., 2017*; *Natusch et al., 2017*). Additionally, captive specimens are less subjected to some of the main factors influencing isotopic fractionation and tissue turnover, such as natural habitat gradients, seasonality, complex food chain, diet quality variation, and nutritional stress (*Shipley & Matich, 2020*).
Several studies have successfully demonstrated the potential application of SIA to differentiate wild from captive animals (*Dittrich, Struck & Rödel, 2017*; *Natusch et al., 2017*; *Brandis et al., 2018*; *Alexander et al., 2019*; *Andersson et al., 2021*; *Hopkins et al., 2022*). However, there is no scientific compilation of the topic. This study examined an extensive collection of publications using SIA in wild and captive animals. The available data in the literature was organized in a database, making the systematized information available for academic and applied purposes. We performed qualitative and quantitative analyses to develop an overall picture of the application of stable isotopes to distinguish between wild and captive animals focusing on: (1) evaluating the potential of this tool to distinguish wild and captive individuals; (2) searching for discernible patterns of how such differences occur.

## SURVEY METHODOLOGY

### Data source and compilation

We searched peer-reviewed publications in the Web of Science database (https://clarivate.com/webofsciencegroup/solutions/web-of-science/) from 1945 to 2021 using the terms "isotop*" AND "wild OR free-rang*" AND "captiv* or farm*" as a topic. We added the search terms "NOT 'chicken OR hen* OR cattle OR pig'" to exclude domestic animals from the results. The search returned 295 hits, initially sorted based on the title, keywords, and abstract. First, we considered studies of stable isotopes involving any nondomestic vertebrate species. In a second instance, we selected only research related to carbon, nitrogen, hydrogen, oxygen, or sulfur isotopes that met one of the following criteria: (1) used stable isotope to differentiate wild from captive animals; (2) conducted isotopic analysis in wild and captive individuals of the same species in the same study. Paleontological publications and those studies exclusively using compound-specific isotope analysis (CSIA) techniques were excluded. We also excluded studies that did not show basic isotopic statistical information, such as average, standard deviation, or error. Data were extracted only from original research papers rather than those found in reviews or meta-analysis studies to avoid duplicates. After these two filtering steps, 47 studies remained to be analyzed in this review (Table S1). We also checked the references list from the main studies and reviews to ensure that all relevant papers were included.

Data were initially collected from the texts and tables of articles. When they were not or were only partially available, we contacted the authors asking for the missing or raw data. As a last resort, we estimated isotopic values from the figures, when available, using PlotDigitizer software, version 2.1.1 (*PlotDigitizer, 2022*).

### Data and metadata structure

We selected variables related to the taxon classification, biology and morphology of animals, samples data (tissue, geographic location, year, and period), rearing system, isotopic records (values of mean, standard deviation, minimum and maximum, range), methodological records from the isotope analyses (analytical error, lipid extraction, and international reference material), and identification of publication were registered (Table 1 and Table S1). Regarding the rearing system, we classified the animals as "wild" or

**Table 1 List and description of the variables selected to be included in the database.**

| Variable | Explanation |
| --- | --- |
| Reference | Publication included in the data collection. |
| Taxon group | Mammal, Bird, Reptile, Amphibian, Fish. |
| Taxon | Most detailed taxon identified (usually species or genus) |
| Life-stage | Adult or subadult |
| Size-range or weight | Body size in centimeters or weight in kilograms |
| Diet | Herbivore, carnivore or omnivore |
| Continent | Where data were collected: Africa, Asia, Europe, Oceania, North America, and South America. |
| Multiple countries? | Yes or no. Were samples collected in more than one country? |
| Country/Region | Country(ies) or subcontinental region where data were collected. |
| Region/city | City, estate, or region within a country. |
| Lat | Latitude (m). UTM system |
| Long | Longitude (m). UTM system |
| Month/period | Month or other information available about samples collection period. |
| Year | Year of samples collection. |
| Tissue | Animal tissue used in the isotopic analysis (*e.g.*, feather, muscle, blood). |
| Sub-tissue | A specific part of a given tissue (*e.g.*, red blood cells, type of feathers). |
| Rearing system | wild or captive |
| Subgroup | When there are different treatments within a wild or captivity condition. |
| $N$ | The number of sampled animals. |
| Rearing system change | Time the animal changed from wild to captive or captive to wild (in months). |
| Mean $\delta^z x$ (‰) | Isotopic ratio means. |
| SD $\delta^z x$ (‰) | Isotopic ratio standard deviation |
| MIN $\delta^z x$ (‰) | Isotopic ratio minimum value |
| MAX $\delta^z x$ (‰) | Isotopic ratio maximum value |
| Range $\delta^z x$ (‰) | Difference between maximum and minimum isotopic ratios |
| Lipid extraction | Yes or no. Were lipids extracted during sample preparation? |
| Analytical error | Error that might be associated with isotope-ratio mass spectrometry |
| Reference standard | Compounds with well-defined isotopic compositions used to ensure accuracy in mass spectrometric measurements of isotope ratios |
| Observation | Any additional relevant information |
| Related publication | DOI or link to the publication |

"captive." On some occasions, the origin of the samples was uncertain; thus, we followed the authors' conclusions about their wild or captive origin. All stable isotope results are expressed in the conventional delta ($\delta$) notation, in units per mil (‰).

We used the systematized metadata of our database to present an overall picture of the studies analyzing stable isotopes in wild and captive animals. Additionally, we used the isotopic ratios' mean, standard deviation, and range of wild and captive animals from the database to evaluate general differences and similarities. For this, we performed Student's

$t$-test for differences in $\delta^{13}$C, $\delta^{15}$N, $\delta^{18}$O, $\delta^{2}$H, and $\delta^{34}$S considering the whole database. We also tested for differences between wild and captive animals for $\delta^{13}$C and $\delta^{15}$N per continent, taxon group, and/or dietary category. Finally, we performed Student's $t$-test or one-way ANOVA to compare $\delta^{13}$C and $\delta^{15}$N in wild and captive animals considering the terrestrial taxon together (amphibian, reptile, bird, and mammal) and fish per continent. Due to the lack of suitable studies, we did not perform more detailed analyses for $\delta^{2}$H, $\delta^{18}$O, and $\delta^{34}$S. In addition, despite the possible effects of different types of captivity on the isotopic signature of animals, we could not access details of captivity conditions in most studies, making it impossible to categorize them as suggested by *Hudson (1989)* or CITES. Thus, in the database, they were called "captivity" only.

To gain a deeper understanding, we also assessed each study individually. For studies that analyzed isotopic differences between the two environments, we relied on the results reported by the authors. For studies that measured but did not compare isotopic ratios in the wild and captivity, we tested for such differences ($t$-test, ANOVA, Wilcox-test, Kruskal-Wallis, and linear mixed model) when the original data were available. In this approach, we considered the different categories of wild or captivity when explicitly presented by the authors. Inferential tests were preceded by analyses of normality (Shapiro-Wilk test) and equality of variances. We performed statistical tests in the R platform, version 4.1.0 (R Development Core Team), with a significance level of 5% in all hypothesis testing.

## Quality assurance and control

As part of the quality assurance, we carefully checked the data in different steps of the database building, trying to keep the information as close as possible to the original one. For example, we double-checked for mean, standard deviation, and range of isotopic ratios extracted by PlotDigitizer. We also double-checked the original source outliers detected by boxplots for the same variables.

When not provided, the geographic coordinates of the samples were estimated based on the authors' most detailed geographic information (*e.g.*, city, region, fishing area zones). Species names were kept as in the original publications, and as "fishes" are a paraphyletic group, we checked the taxonomic class of each species using Eschmeyer's Catalog of Fishes (*van der Laan & Fricke, 2023*). Species diet classifications were checked using the R package Sider (*Healy et al., 2018*) or searched for in peer-reviewed papers.

## RESULTS

### An overall picture of the use of stable isotopes to differentiate wild and captive animals

The first studies using stable isotopes to distinguish wild and captive animals dated around two decades ago and aimed to evaluate the potential of SIA as a tool to differentiate wild from recent farm-escaped salmon (*Salar salar*, *Dempson & Power, 2004*) and mink (*Mustela vison*, *Hammershøj, Asferg & Kristensen, 2004*). Since then, the number of studies using this tool and its applications has been growing.

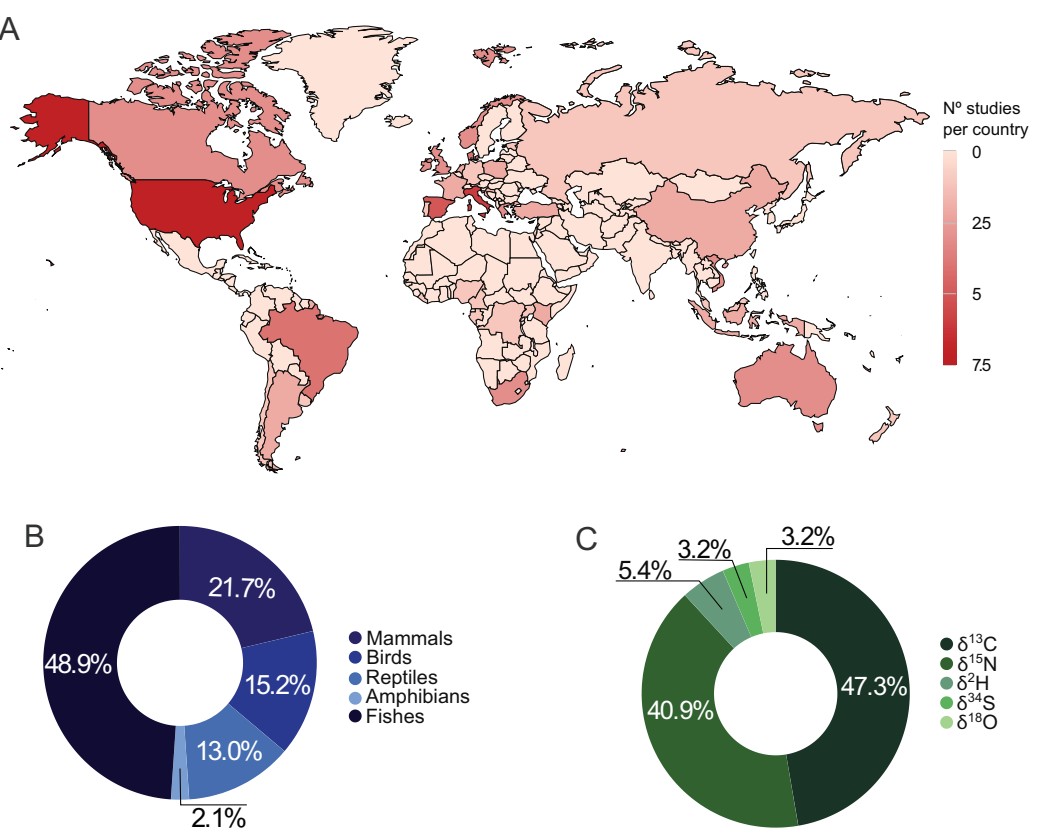

**Figure 1 Distribution of studies using stable isotopes of carbon, nitrogen, hydrogen, oxygen, and sulfur in wild and captive animals worldwide (A), by taxonomic group (B), and by isotope element (C).**

From the 47 selected publications, 33 used stable isotopes to distinguish wild from captive animals (Table S2), and 14 analyzed stable isotopes in wild and captive vertebrates for different purposes (Table S3). We found studies distributed in 37 countries (Fig. 1A), with the United States and Italy having more publications ($n = 6$ each). Regarding the continents, Europe ($n = 21$) and Africa ($n = 3$) had the largest and smallest number of studies, respectively.

Fifty-five species from different vertebrate taxonomic groups (mammals, birds, reptiles, amphibians, and fishes) were studied. The most studied groups varied on different continents: while studies on reptiles and amphibians were concentrated in Asia and Oceania, studies with fish were distributed worldwide (Fig. S1). All studies focused on one species ($n = 36$) or a group of species from the same taxonomic class ($n = 11$). Amphibia was the least representative group, recording in only one study, while fish accounted for over 46% of the publications (Fig. 1B).

Carbon and nitrogen stable isotopes in combination accounted for 80.8% of the studies, and 88.2% used at least one of those elements (Fig. 1C). Hydrogen, oxygen, and sulfur stable isotopes together account for less than 15% of the publications (Fig. 1C). Muscle and inert organic structures were the most analyzed tissues presenting in 46.81% and 42.55% of

**Table 2 Comparison of the mean isotopic ratios, standard deviation, and range of $\delta^{13}$C, $\delta^{15}$N, $\delta^2$H, $\delta^{18}$O, and $\delta^{34}$S in captive and wild animals considering all 47 analyzed publications.**

| | $\delta^{13}$C | $\delta^{15}$N | $\delta^2$H | $\delta^{18}$O | $\delta^{34}$S |
|---|---|---|---|---|---|
| $\mu_w$ | −20.42 ± 4.13[a] | 10.92 ± 4.20[a] | −68.80 ± 33.90[a] | 23.20 ± 1.83[a] | 1.50 ± 8.74[a] |
| $\mu_c$ | −19.68 ± 3.09[a] | 10.18 ± 3.45[a] | −61.21 ± 38.89[a] | 19.05 ± 1.66[b] | 8.16 ± 7.28[a] |
| $SD_w$ | 0.90 ± 0.63[a] | 0.86 ± 0.70[a] | 10.35 ± 3.70[a] | 1.89 ± 0.48[a] | 2.17 ± 2.32[a] |
| $SD_c$ | 0.68 ± 0.61[b] | 0.56 ± 0.53[b] | 6.73 ± 5.71[a] | 1.44 ± 0.50[a] | 1.24 ± 2.34[a] |
| $Range_w$ | 3.31 ± 2.35[a] | 3.50 ± 2.53[a] | 37.02 ± 24.49[a] | 7.3 ± 2.24[a] | 6.24 ± 6.48[a] |
| $Range_c$ | 2.48 ± 1.98[b] | 2.04 ± 1.70[b] | 29.29 ± 29.64[a] | 6.02 ± 2.46[a] | 7.92 ± 9.73[a] |

**Note:**
Significant differences are indicated by different letters ($p < 0.05$).

the studies, respectively. Muscle was used in 83.33% of the works with fishes, while inert tissues were used in 86.96% of studies involving the other taxonomic groups.

The experimental design, captivity condition, and description of the publications varied widely (Table S4). While some studies measured stable isotopes for only two categories (wild and captive) of the same species at the same moment and in the same geographic regions, others had a complex design, considering several types of wild or captivity various species, countries, period, and samples origin.

## Patterns and potential of SIA in distinguishing between wild and captive animals

### Analyses of the systematized database

The mean for $\delta^{18}$O and the standard deviation and range for $\delta^{13}$C and $\delta^{15}$N were significantly higher in the wild than in captive animals (Table 2). $\delta^2$H, $\delta^{18}$O, and $\delta^{34}$S standard deviation tended to be higher in wild animals, but it was not significant (Table 2). The isotopic differences in the rearing system varied with the geographic location, taxonomic group, and diet (Fig. S2).

$\delta^{13}$C of the terrestrial taxon (amphibian, reptile, bird, and mammal) were significantly higher in captive animals ($t_{121} = 2.40$, $p = 0.02$). At the same time, the $\delta^{15}$N was higher in the wild terrestrial animals but only when controlling for diet type ($F_{1,1} = 363.4$, $p = 0.03$). Captive fishes exhibited significantly lower $\delta^{13}$C in Europe ($t_{37} = −2.07$; $p = 0.05$). The isotopic space occupied by individuals considering C and N simultaneously tended to diverge in all taxonomic groups, either at the mean position or range (Fig. S3).

### Analysis of the publications individually

To identify differences between wild and captive individuals, the studies performed graphical analyses of overlapping, discriminant tests, frequentist statistics (such as *t*-tests and ANOVA), or the combination of the last two statistical methods. Overall, 83.9% of the studies found significant differences or no overlapping among categories of wild and captive animals analyzed, while 16.1% distinguished among some modalities where authors considered different wild or captive conditions. In addition, no publication failed to differentiate between all categories of wild and captivity (Table S2). Accurately

## Quantitative comparison between environments

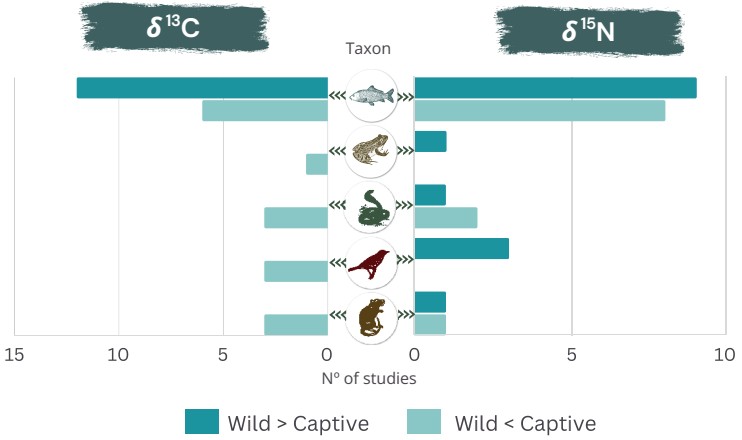

**Figure 2 Quantitative comparison (higher or lower) of $\delta^{13}C$ (left) and $\delta^{15}N$ (right) between wild and captive animals by taxonomic group.** Data extracted from each publication individually.

identifying the group to which individuals belonged (wild or captive) ranged from 58% to 100% when using discriminant tests.

From the studies using $\delta^{13}C$ and $\delta^{15}N$, 90.3% and 89.3% reported significant differences between wild and captive animals, respectively. The $\delta^{13}C$ was usually higher in the wild than in captive fishes. The opposite was found for the other taxonomic groups (Fig. 2). $\delta^{15}N$ was consistently higher in wild birds and did not show clear trends in the other taxonomic groups (amphibians could not be evaluated since there was only one study) (Fig. 2). The studies used $\delta^{2}H$ and $\delta^{18}O$ when geographic variation was also involved ($\delta^{2}H$ in birds and reptiles and the $\delta^{18}O$ in fishes and amphibians). The $\delta^{34}S$ was used in two studies with birds or fishes, involving expected differences in the proportion of marine/terrestrial diet (Table S2).

Regarding the publications that measured but did not compare stable isotopes in wild and captive animals, we accessed the original data for 10 of 14 studies. We could distinguish between all (64.3%) or some (21.4%) categories in which authors used different wild or captive conditions. We found no differences between treatments in 14.3% of the publications (Table S5).

# DISCUSSION

## An overall picture of the use of stable isotopes to differentiate wild and captive animals

The development and understanding of methods to accurately identify the origin of an animal are crucial to ensure that wildlife farming fulfills its role as a conservation strategy (*Phelps, Carrasco & Webb, 2014*; *Tensen, 2016*; *Lyons, Natusch & Jenkins, 2017*; *Natusch, 2018*). Stable isotope analyses are an important biomarker of animal provenance. Here we summarize the main aspects of the use of stable isotopes as a tool to differentiate wild and captive animals.

Fish was the most studied group involving taxa of a single class (Actinopteri) and mainly species used for human consumption (*e.g.*, European seabass, meagre, and different species of salmon). Only two fish species were considered globally threatened by IUCN and included in the CITES appendix (Tables S3). These findings suggest that beyond environmental impacts, identifying seafood origin involves concerns related to food safety, leading to labeling regulation about the rearing system in European Union and the United States (*Commission Regulation (EC), 2001*; *Agricultural Marketing Service (AMS), 2009*). However, few countries have clear requirements for the origin of the breeding system yet (*El Sheikha & Xu, 2017*).

Mammals, birds, and reptiles studies were mainly associated with ecological or forensic purposes, such as identifying the origin of a wolf population (*Canis lupus*) in an area where they were previously extinct (*Kays & Feranec, 2011*), the use of stable isotopes to identify the provenance of invasive alien species (*Trachemis scripta*) (*Hill et al., 2020*), or to detect crocodile lizard (*Shinisaurus crocodilurus*), short-beak echidnas (*Tachyglossus aculeatus*), and yellow-crested cockatoos (*Cacatua sulphurea*) laundering (*van Schingen et al., 2016*; *Brandis et al., 2018*; *Andersson et al., 2021*). Most studies in mammals, birds, or reptiles involved living species listed in the CITES appendices and relied on inert keratinous tissues, demonstrating SIA as a noninvasive biomarker of the rearing system.

Around 90% of the studies relied on $\delta^{13}C$ and $\delta^{15}N$ in animal tissues, which was not a surprise, considering the assumption that wild and captive animals have different diets, which is reflected in their isotopic ratios (*e.g.*, *Dempson & Power, 2004*; *Natusch et al., 2017*; *Hill et al., 2020*). The $\delta^2H$ and $\delta^{18}O$ have been largely used to infer animals' geographic origin and movement due to their pattern of variation in response to hydrological processes and the linkage with those in animal tissues (*Hobson & Wassenaar, 2019*). However, the tissue-environment relationship of $\delta^2H$ and $\delta^{18}O$ may also be affected by local factors, such as food-web relationships (*Vander Zanden et al., 2016*), physiology, and the proportion of water in the animals' diet (*Magozzi et al., 2019*). The lack of dietary and trophic studies using $\delta^2H$ and $\delta^{18}O$ is also reflected in the wild-*versus*-captive studies. No research used $\delta^{34}S$ exclusively to distinguish wild from captive animals.

## Patterns and potential of SIA in distinguishing between wild and captive animals

### Analyses of the systematized database

We observed significant differences in the $\delta^{18}O$ between wild and captive animals. These findings suggest that $\delta^2H$ and $\delta^{18}O$ may be underused in differentiating wild and captive animals since these elements represented less than 10% of the isotopes analyzed. The $\delta^{13}C$ and $\delta^{15}N$ range and standard deviations were significantly higher in the wild compared to captivity supporting the assumption that wild conditions tend to be more variable than captive ones, regardless of the specific circumstances. Few studies mention data dispersion variables to infer animals' origin (*Molkentin et al., 2007*; *Busetto et al., 2008*; *van Schingen et al., 2016*; *Dittrich, Struck & Rödel, 2017*). However, our results suggest those variables could be relevant in differentiating wild and captive animals, and particularly helpful in identifying the rearing system of a group instead of one particular individual.

Although most studies could distinguish isotopically wild from captive animals (see below), there were few differences when considering the entire database simultaneously. Some patterns emerged as analyses were performed on more homogeneous groups (*e.g.*, only terrestrial taxon or fish in Europe), suggesting that the isotopic differences between wild and captive animals are not unidirectional. Rather, such differences appear to vary by location and taxonomic group.

### Analysis of the publications individually

More than 80% of the publications that looked for isotopic differences between wild and captive animals were successful. In some research, the animals' origin was inferred by the authors or informed by traders or labels, which may have influenced the results. The studies with the lowest performance involved fish (*Pereira et al., 2019*; *Molkentin et al., 2007*, *2015*; *Wang et al., 2018*; *Vasconi et al., 2019*; *Liu et al., 2020*). The complex experimental design, the largest number of variables involved (including different fish farming models) (Table S3), and the uncertainty of the samples' origin were probably the main reasons this group performed worse than terrestrial taxon. Conversely, other studies could isotopically distinguish animals from different rearing systems at even more detailed levels, such as distinguishing different breeders of the same species (*Castelli & Reed, 2017*).

Despite the specifics of each study, $\delta^{13}C$ was consistently lower in the wild than in captive terrestrial animals (Fig. 2), indicating consumption of higher levels of C4 plant-based food for captive individuals compared to wild populations of the same taxon. Such a pattern indicates the composition of industrial food provided to these captive animals, based mainly on less expensive items, such as corn (*Kays & Feranec, 2011*). In contrast, wild fishes exhibited higher $\delta^{13}C$ than captive ones in most studies. Multiple factors can explain these findings, such as captive fishes tend to have higher lipid concentrations than wild ones, leading to lower $\delta^{13}C$ (*DeNiro & Epstein, 1977*; *Focken & Becker, 1998*; *Serrano, Blanes & Orero, 2007*; *Fasolato et al., 2010*), most fish studies included carnivore marine or migratory species in Europe, where the composition of conventional aquafeeds is based on terrestrial plant ingredients such as cereals, soy, legumes, and plant-derived oils, which has lower $\delta^{13}C$ than fish-based diet typical of carnivore wild marine fishes (*Schoeninger & DeNiro, 1984*; *Farabegoli et al., 2018*; *Wang et al., 2018*).

Finally, we found differences between wild and captive animals in more than 85% of the studies that did not compare individuals in these two environments. These results suggest the potential of isotopes to differentiate wild and captive animals even when the research was not designed to look for such differences. They are also particularly relevant, considering the publication bias of positive results (*Mlinarić, Horvat & Šupak Smolčić, 2017*), which could overestimate the capacity of SIA in identifying animals' origin. The only two studies that found no differences relied exclusively on $\delta^{13}C$ and had high variability and unbalanced samples (*Cree et al., 1999*; *Hammershøj, Asferg & Kristensen, 2004*). Using more than one isotopic element and more sensitive or complex statistical tests (*e.g.*, multivariate analysis) could help find masked differences between wild and captive groups.

***Improving isotopic research to distinguish between wild and captive animals***

Recognizing the differences in the experimental design and objectives of each study, some approaches can be considered to enhance and improve the use of stable isotopes to distinguish between wild and captive animals. First and crucial, the study should define known captive or wild origin samples for comparison whenever possible, whether directly sampled or from the literature.

Second, the choice of the tissue used in the research is fundamental. Metabolically active tissues reflect distinct temporal integration time according to their turnover rate, varying from a few days (*e.g.*, blood plasma), months (*e.g.*, muscle), or lifetime (*e.g.*, bone collagen) (*Tieszen et al., 1983*; *Vander Zanden et al., 2015*; *Carter, Bauchinger & McWilliams, 2019*). Conversely, the isotopic ratio of keratinous tissues, such as feathers, claws, and scales, will reflect where they were synthesized since they are metabolically inert after their formation (*Mizutani et al., 1990*; *Hobson & Clark, 1992*). Therefore, under changes between rearing systems, the study should identify the exact period of change to avoid any confounding effect of feeding changes and consider the tissue that captures the time elapsed since the rearing change.

Third, to improve the evaluation and understanding of the data, special attention must be given to presenting the methodology and results. Measures of central tendency, such as each treatment's mean and standard deviation, should always be reported. The wild and captive conditions should also be carefully considered and described to ensure correct interpretation of the results.

Despite increased publications using stable isotopes as tracers of wildlife origin in recent decades, there are still some important gaps that future studies could address. Some taxon groups are highly underrepresented, especially amphibians, with only one study. The influence of different categories of captivity on isotope ratios also needs to be better explored and understood. Most studies are in Europe and North America, while some high diversity and threatened regions in South America and Africa have been little studied. Regarding the isotopes, $\delta^{18}O$, $\delta^2H$, and $\delta^{34}S$ were only occasionally analyzed, and the potential of $\delta^{18}O$ and $\delta^2H$ to distinguish wild and captive animals based on differences in physiological conditions in these environments remains unexplored.

## CONCLUSIONS

Our study reveals that SIA can help distinguish between wild and captive origin in different vertebrate groups, rearing conditions, and methodological designs. Despite the variety of publications reviewed, we could observe some distinct patterns in how these differences occur, such as the higher diet variability in wild animals and the preferential use of plant-based food in captivity.

Nevertheless, some aspects should be carefully considered for the proper use of the methodology, such as knowing the wild and captive conditions of the animals studied and having samples of known origin to use as a basis for comparison. Additionally, the methodology seems to perform better the more homogeneous samples are since the direction of differences between wild and captive animals can vary greatly according to local and taxonomic specificities.

Many gaps remain to be filled, especially in the unbalanced taxon, region, and isotope studied. We expect the present study to expand the use and acceptance of SIA as a reliable tool in identifying animals' rearing system origin and, consequently, contributing to the efficiency of wildlife farming as a conservation strategy and protecting natural populations.

## ACKNOWLEDGEMENTS

We thank the Environmental Isotope Studies group at the University of Brasilia for all scientific discussions and contributions. Jim Hesson copyedited the manuscript (https://www.academicenglishsolutions.com).

### Funding

This work was supported by the University of Brasilia (Process n° 23106.033190/2021-00). The funders had no role in study design, data collection and analysis, decision to publish, or preparation of the manuscript.

### Grant Disclosures

The following grant information was disclosed by the authors:
University of Brasilia: 23106.033190/2021-00.

### Competing Interests

The authors declare that they have no competing interests.

### Author Contributions

- Luiza Brasileiro conceived and designed the experiments, performed the experiments, analyzed the data, prepared figures and/or tables, authored or reviewed drafts of the article, and approved the final draft.
- Rodrigo Ribeiro Mayrink conceived and designed the experiments, performed the experiments, authored or reviewed drafts of the article, and approved the final draft.
- André Costa Pereira performed the experiments, analyzed the data, prepared figures and/or tables, authored or reviewed drafts of the article, and approved the final draft.
- Fabio José Viana Costa analyzed the data, authored or reviewed drafts of the article, and approved the final draft.
- Gabriela Bielefeld Nardoto conceived and designed the experiments, authored or reviewed drafts of the article, and approved the final draft.

### Data Availability

The raw measurements are available in the Supplemental Files.

The data is available at Mendeley Data: Environmental Isotope Studies, EIS (2022), "Studies using stable isotopes to differentiate wild and captive animals", Mendeley Data, V1, DOI 10.17632/vt9jf3hz6h.1.

## Supplemental Information

Supplemental information for this article can be found online at http://dx.doi.org/10.7717/peerj.16460#supplemental-information.

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
