# Peer review of "Differentiating wild from captive animals: an isotopic approach"

_PeerJ, doi:10.7717/peerj.16460_

## Round 0.1 · original submission · Major Revisions

Your manuscript requires extensive revisions. The two reviewers have provided you with thorough and thoughtful reviews. Please pay close attention to the details during your revisions to ensure your re-submission is acceptable.

Reviewer 1 ·

Basic reporting

The role of captive wildlife in conservation and the identification of material sourced in wildlife trade is a complex topic at which the authors have made a decent attempt to review. However, in its current state I cannot recommend the manuscript for publication. My specific concerns are around the framing of the manuscript and the chosen focus, and analyses, for what is a potentially useful dataset. This manuscript currently comes across as confused in its position between being a meta-analysis and a review. My primary contention is that the analyses, which focus on averaging isotopic ratios, do not usefully add information to the field of study. The authors themselves acknowledge significant variation between species, geographic location, tissue types, etc., yet proceed to look at averages over taxonomic and geographical scales that render the results of little utility. Rather than focus on averages of isotopic levels, which have little purpose, this review could usefully focus on evaluation of conditions under which captive and wild differentiation has been successful and provide recommendations based on the literature search. An additional concern is that there is insufficient acknowledgement of the complexity and diversity of captive farming types – which have potential to greatly influence isotopic compositions. The authors have chosen an ambitious subject to review all studies on this topic, however this review can only be correctly made by explicitly considering the different degrees of captivity which exist, for example: wildlife may be ranched, captured from the wild and raised, intensively farmed, parent reared, surrogate reared etc. I would also hope to see greater consideration of the complex role that captive bred wildlife plays in conservation. I appreciate the author’s position as a law enforcement practitioner (as I am) and note the difference between the content of the cover letter and the content of the manuscript. The manuscript opens with a purported focus on law enforcement, however there is scant mention of this onwards in the manuscript and the dataset is not explained in terms of law enforcement utility – I feel that this law enforcement focus would needlessly constrain the review if it were to be adequately addressed. Nevertheless, there are useful points to be raised about the utility of isotopic analysis in wildlife law enforcement, but, as mentioned, a greater depth of background and review of literature, including discussion of forensic admissibility, would be required.
The manuscript at his stage lacks clearly articulated aims and objectives of the review and is too general to bring insight to the field. The paper could more usefully focus in more detail on successes and failures of isotopic analysis to differentiate between captive and wild along with more consideration of the contextual complexities involved in isotopic analysis.
My suggestion is to reject and resubmit with major changes.

Experimental design

see previous

Validity of the findings

see previous

Additional comments

Specific comments:
Line 60: I am unsure of the opening with specific focus on law enforcement with little further mention of relevance onwards in the review.

Line 86: There needs to be a section of farming of wildlife/at least some acknowledgement of the multiple reasons why wildlife is kept in captivity.
Line 98: Need more on various types of laundering and perhaps examples of when laundering is perceived to be a problem.
Note that farmed wildlife in law enforcement and conservation is complex and requires a more developed background.
Line 91: “are usually inaccurate and easily defrauded” – this assertion is uncited and in many cases wrong.
Line 98: This is not necessarily a more recent application than the previous examples.
Line 102-104: Note that these were not real legal cases rather they were research projects and ae unlikely to have been used in legal cases – the distinction between research/intelligence and legally robust forensics is important and often missed in academic publications, see Ogden, R., Dawnay, N., & McEwing, R. (2009). Wildlife DNA forensics—bridging the gap between conservation genetics and law enforcement. Endangered Species Research, 9(3), 179-195.

Line 106-107: See previous
Lines 109-118: this section needs better/ more complete citation
Line 121: Note there are other reasons
Line 123-125: Not necessarily true. Natusch is talking about a specific instance with snakes, single cases cannot be extrapolated as generalisations (a key finding of this paper!)
Line 125-127: “Wild animals are also more susceptible to nutritional and water stress, which can influence their tissue isotopic ratios (Doi, Akamatsu & Gonzalez, 2017; Magozzi et al., 2019).” I question whether this is strictly true. The citations do not robustly support the statement.
Line 134: I’d like details on the specific questions to be addressed by this review/meta analysis.
Line 136: I’m perplexed by the focus on law enforcement which needlessly constrains the scope of the review. There is also no discussion of the specific requirements of forensically robust law enforcement tools in the paper, neither do the authors make use of their dataset to explore which published studies have law enforcement relevance.
Again, see Ogden, R., Dawnay, N., & McEwing, R. (2009). Wildlife DNA forensics—bridging the gap between conservation genetics and law enforcement. Endangered Species Research, 9(3), 179-195.
For further details on wildlife law enforcement in a forensic context.
Line 169-170: This is a big assumption about presumed wild and presumed captive, why not separate the designations?
Line 186-187: I’d like full details on the parameters of this qualitative analysis.
Line 189: As mentioned in the overall comments I think more useful would be showing whether studies could or could not fully differentiate between wild and captive.
Line 203: “Most of them used stable isotopes” Given the focus of this paper I would expect all of the analyzed publications to do this!
Line 206-207: I’d like to see the specific numbers or %
Line 214: When you say “most studies” please include specific numbers or %
Line 217-218: This is well accepted as the usefulness of these isotopes.
Line 224: Again when saying “most” I’d like numbers quoted in the text for clarity.
Line 227-231: I struggle to see the purpose in these mean isotopic analyses. Given well acknowledged variations between species and tissue types in isotopic compositions there is little to nothing to be gained from large scale mean calculations. I suggest they be removed.
Line 237-238: “varying with the geographic location and taxonomic group” This is unsurprising. Again I struggle to see the relevance of these large scale means given the well acknowledged geographical variance in stable isotope rations. The use of large scale means is further inappropriate because the authors give no consideration to the modes of captive husbandry in this paper. Modes of husbandry are not discussed in this paper and appear to be considered as uniform – which they are not, and hold the potential to drastically influence whether captive and wild specimens can be differentiated.
Line 248: Similar point to the above RE large scale means within taxonomic groups.
Line 266: changes in spelling: Farmed; salmon; mink
Line 270: What did the other studies do?
Line 274: I’d like to see numbers or % in the text here.
Line 281: About 87%? I think this needs specificity as it is a key point of the paper. I think it would also be useful to provide a break-down of significant results by taxonomic groups instead of the relatively meaningless large scale isotopic value
Line 285-286: This is the first mention of sub-divisions of captivity type – this needs more extensive exploration in the introduction and I suggest that the meta-analysis take into account the specific characteristics of captivity from the start.
Line 290-291: Please provide details.
Lines 304-305: Citations?
Line 307: What constitutes ‘partial’ success is unclear – could also mean failure.
Line 308: I think this is because in this study there has been insufficient consideration of the variables which contribute to isotopic difference in organism tissue.
Line 310-312: I argue that this was obvious from existing literature and could have been summarized in the introduction – this finding is of insufficient novelty.
Line 317: I’d like to see these hypotheses made a-priori in the text and backed up with specific numbers from the meta analysis.
Line 317-320: The meaning here is unclear.
Line 320-321: I think this could be usefully explored in the data that the authors present.
Line 324: As previous comment.
Line 327: This review could usefully make recommendations for the effective use of stable isotopes in differentiation of sources, but requires supporting evidence from the meta analysis.
Line 331: citation
Line 354: As does a range of previously published literature – again question the novelty of this study and whether it is usefully contributing information.
Line 361: I’d like to see more formal analysis of these differences in captivity across the meta analysis dataset.
Line 410: There has been an attempt to focus on animal trade but the information relating specifically to trade and discussion of law enforcement is vague throughout. The manuscript could be significantly improved by making the decision to focus on reviewing the utility of isotopes for determining between captive and wild sources, or evaluating the potential for such techniques in law enforcement contexts.
Line 411-412: I find it unclear exactly how this review has advanced understanding? – these advances should be set out specifically.
Line 414: Reliability and robustness is important for law enforcement applications. SIA cannot, without specific analysis, be said to be reliable – I note that this paper does not test or evaluate the reliability of methods used and results gained in the contributing papers, thus this statement is not evidenced by the body of the manuscript.
Line 415-416: Like the above the paper does not provide a review of ex situ conservation or population protection – I struggle to see how this statement concludes the paper.

Annotated reviews are not available for download in order to protect the identity of reviewers who chose to remain anonymous.

Reviewer 2 ·

Basic reporting

This is a timely review on a relatively new and developing topic - the use of stable isotope analysis in different fields to determine whether an animal is captive bred or from the wild.

The paper is structured well and is clear to read, although there are some grammatical errors throughout that cause some ambiguity in places (see specific comments below). The manuscript would benefit from more reference to general SIA literature, particularly to support interpretation of results in the discussion section.

The introduction covers many key topics, but is a bit unfocussed as written and so some of the strength of the argument is lost. I would suggest that the authors carefully revise the introduction to more tightly focus on the issue of determining captive vs. wild: why is it important, how is it important to the different fields mentioned (commercial farming/breeding, wildlife trade, ecological impacts of introduced species), how can SIA contribute to determining the rearing status of the species. It is a bit confusing to combine commercial interests with wildlife trade as these are two very different issues, and the authors could strengthen the introduction by clearly introducing the similarities and differences between them (with regard to the need for understanding the captive/wild status of an individual). The literature on SIA in ecology is huge and I think it is distracting to include mention of the broad ecological use of SIA in this paper, at least more than breifly.

I would suggest combining the paragraphs on Lines 88-109 and explaining more clearly in this paragraph on why being able to differentiate between captive and wild is important in the various contexts presented (e.g. forensics, commercial, introduced species). All of these contexts have different requirements and it would help to differentiate between them. I would also suggest removing content not directly related to this question (e.g. the content on ex-situ conservation), and expand some of the background explaining the situations where understanding the rearing history of an individual would be important, the utility of SIA to answer those questions, and a clear explanation of how you would expect isotope ratios to vary on the scale examined in the manuscript. Are there important differences between these two applications?

The discussion would benefit from making broader reference to the literature on SIA outside the studies reviewed to support key points. For example, a clearer explanation of the importance of considering turnover rates in different tissues is needed, with reference to literature to support.

Experimental design

The authors identified 47 papers that use SIA to determine the captive/wild status of individuals for varying purposes, this is a good number and a thorough review at this point could really help to summarize what we have learned so far and highlight what the major knowledge gaps and needs for future research are. This manuscript makes a good start towards that, but could with the data already on hand go even further to provide a really comprehensive review of the field.

While the literature methods are well explained, the methods for the statistical analyses are not explained in enough detail to be able to understand why each test was done. For me one of the major questions is why the analysis was conducted by pooling all of the data from the 47 papers for a re-analysis. Given the variation that exists in stable isotope ratios between taxa, geographic region, temperature etc., I don't see what is gained by taking this approach. It doesn't particularly surprise me that there were few general patterns found given this, but I don't think this in any way means that SIA is not useful, but I think that different analyses could be done with the data gathered. If the authors do have reason to expect some general patterns, then there needs to be more information in the introduction and methods to justify this expectation. The authors also need to build a case for why they might expect a consistent pattern in mean and variance in stable isotope ratios across all studies.

To me the most interesting aspect of this review would be an understanding of what proportion of studies were successful at differentiating between captive and wild individuals – I would suggest the authors expand on this section and explore this aspect of the study more. What made those studies successful? Why were the others not successful? Were some taxa easier to distinguish than others? What can we learn from reviewing these studies and what further work remains to be done?
It is less clear why it is interesting to break down studies by country as there is no clear reason why results should vary by country. The authors provide data on mean isotope values for different dietary guilds (carnivores, herbivores, etc) – I would suggest the authors expand on this to explore how this relates to whether studies were successful in distinguishing between captive/wild individuals. Would we expect consistent variation across these groups? Why or why not?

Figures:
Figure 3 could use clearer labelling - on the version I have the panels are not labelled A-D, although they are referred to in this way in the legend. The legend should also explain what the box and whiskers represent, and indicate statistical significance where applicable.

Validity of the findings

I would suggest to restructure the results to focus primarily on the captive vs. wild results as this is the main question posed in this paper. The section on general results is currently much longer and more detailed than the section on captive vs. wild studies. I would suggest shortening the general results section and then expanding on results related to captive vs. wild studies. The conclusion could expand on unresolved questions and provide guidance for future directions. This would involve engaging more with the literature on SIA to explain factors leading to variation in measured SI ratios, etc. Specific comments below.

Additional comments

Introduction: Are there important differences between the different context in which understanding captive/wild is important? E.g. are there different requirements for accuracy between commercial and forensic applications?
Line 91: I wouldn’t say that the traditional control techniques are “usually” inaccurate without any data or references to back that up. I would suggest replacing “usually” with something like “can be inaccurate”.
Line 93: Expand on this mention of how the knowledge of the rearing history of an individual helps with an understanding of potential invasive populations – perhaps illustrating with an example from the literature.
Line 106: Expand on the explanation of this study – why did the authors conclude that SIA is the best tool for identifying captive/wild individuals, based on what results/findings? This will help the reader to more fully understand the potential advantage of SIA in this context.
Lines 109-122: This is an important section of the introduction as it will provide important knowledge of how stable isotope ratios are expected to vary in the context of the current study. I would review this section and add in additional information that explains more about what factors lead to variation in stable isotope ratios, both due to diet and geographic location.
Line 123: The diet of individuals held in captivity likely varies depending on many factors and may not always be consistent. Diet provides by private owners for example is likely to vary more than diet provided by some commercial enterprises. I would either support this statement with references from the literature, or rewrite to provide a clearer explanation of why diets in captivity would be expected to vary from diets in wild animals.
Lines 180-184: The methods here on the statistical analyses need to be expanded to describe more clearly how the analyses were done. Either here or in the introduction, it would help to provide some justification for the analyses performed. As I mentioned above, I do not see what is gained by pooling the studies and looking for a consistent pattern in the mean and variation in SI rations across all studies as there will be so much variation depending on species, diet, geographic locations etc. If there is a good reason for doing this, the authors can provide more explanation to explain what is gained by pooling studies in this way. However I would suggest that the authors consider analysing the data in a different way that is more relevant here – perhaps comparing effect sizes across studies to see how robust these types of studies are in being able to differentiate between captive vs. wild.
Line 205: It is not clear what is meant by “analyzed stable isotopes in wild or captive vertebrates focused on dietary analysis” – how is this different than the other studies? It would help to clearly define the different types of studies somewhere (intro or here) and then use consistent terminology throughout.
Line 220: The word “heterogeneously” is used a few times in the manuscript where I think a simpler term would be clearer. Here I think you mean to say that there is variation on what regions studies which taxa – I would expand a little here to more clearly explain this.
Lines 231-240: This would fit better in the intro or discussion as it is interpretation of results.
Line 284: "A couple of studies showed some overlap between wild and captive samples" - what is meant by "some overlap" here - overlap in what specifically?
Line 288: Define how semi-captive differed from captive and wild
Line 307: Explain what is meant by “partially” succeeded – how is this defined?
Line 309: It is stated here that the study did not find “a general trend in how this differentiation occurs” - as mentioned above, it should be explained here why this was expected, and the reasons for why such a pattern were not found should be discussed
Line 335: This paragraph talks about lipid extractions, which have not been mentioned previously in the paper. A clear explanation of why this procedure is done and what the impacts of lipids on the isotope ratio measurements would be.

---

## Round 0.2 · Major Revisions

Both reviewers have made substantive comments on both of the iterations of this manuscript, and R1 notes that the manuscript is improved at this point. I also note that the authors' rebuttal was very substantive and complete. I was vacillating between a major and a minor revisions decision, but considering the number of comments from both reviewers in this iteration, I think that a major revisions decision is warranted.

Note in particular –

from R1: "...there is still a tendency to explore and discuss the content of individual papers that I feel is largely unnecessary as it leads to a discussion that could be made more concise."

from R2: "While the review covers most of the key papers in this area, there is a lack of critical analysis that is needed to move this beyond a summary of the state of the field."

"More detailed methods would help to understand what specific questions were being addressed and how each analysis was conducted..."

I expect that both reviewers will be fine seeing a revision of this MS, as befits a major revisions decision. Thank you to both of the reviewers for their work on this so far, and to the authors for their comprehensive revisions/rebuttal.

Reviewer 1 ·

Basic reporting

see additional

Experimental design

see additional

Validity of the findings

see additional

Additional comments

I think this is a greatly improved manuscript, well done. The tighter focus makes the key point stand out better and has vastly improved the readability of the manuscript. That said, there is still a tendency to explore and discuss the content of individual papers that I feel is largely unnecessary as it leads to a discussion that could be made more concise. I appreciate that authors may feel that they have to generate papers of a certain length, but I would prefer that this paper conveyed the point clearly.
As a rule, only discuss in the Discussion section what is detailed in the Results section. There are several instances when specifics are introduced in the Discussion that are not presented in the results – there is no need for this and it creates confusion and unnecessary work for the authors.
The new Figures are good but need captions and axis labels to be usable.
I am unconvinced of the necessity to use Group 2 in this publication. In order to be convinced I would like to see more clarity and explanation around how the authors carried out their own analyses on the data in papers that did not explicitly seek to differentiate wild from captive. As the authors are generating new results this needs to be clearly described in the methods and documented as a supplementary dataset – I suggest a table that includes the statistical analysis used. In all tables it would help if the authors indicated whether the reference used is Group 1 or Group 2.
The results discussion could be more clearly organised by using headings and sub-headings in addition to Group 1 and Group 2. I think that sub dividing by taxon would be the most logical and would allow you to clearly present and discuss them. At present it is a bit confusing and focuses a lot of discussion on fishes to the detriment of other taxa.
An additional minor point is about language: I have made some grammatical and phrasing correction suggestions throughout. They are not exhaustive, and the authors may wish to use an English language proof-reader. However, they have done far better than I could do if I had written a manuscript in Portuguese and the meaning of the text is clear. Ultimately, I believe this is an issue for the journal to decide on and, in my view, should not preclude publication.
I think that the corrections probably sit on the border of major-minor. That said I think that they are more an issue of level of description in the Methods and clarity of layout in the Results and Discussion than any conceptual or content problem. If the authors pay close attention to a logical structure for organising the Results and Discussion then I see no reason why this manuscript should not be published.
Some specific comments on the manuscript (NB the line references refer to the TRACK CHANGES document):
L117: overexploiting -> overexploitation
L118: captive -> in captivity
L128: “born in captive”, “bred in captive” -> captivity
L140: Easier and clearer to say 1970s 1980s
L140-L141: Internationally traded wildlife was based mainly on wild sources
L142: Nowadays -> in recent times
L143-144: Being simple and uncontentious conservation strategies
L146: It has been suggested that wildlife farming should meet…
Also consider citing different publications instead of relying too much on Tensen:
Phelps, J., Carrasco, L. R., & Webb, E. L. (2014). A framework for assessing supply‐side wildlife conservation. Conservation Biology, 28(1), 244-257.
Challender, D. W., Ades, G. W., Chin, J. S., Sun, N. C. M., lian Chong, J., Connelly, E., ... & Nash, H. C. (2019). Evaluating the feasibility of pangolin farming and its potential conservation impact. Global Ecology and Conservation, 20, e00714.
L150: Typo – damage
L153: Are not -> may not
L159: I don't think develop new techniques is quite right - perhaps better to say use robust techniques. After all, stable isotope analysis is not a really new technique - what you are arguing here is that it exists and should be made better use of for regulation
L181-182: This sentence is a little clumsy, perhaps better just to say that C, N, H, O and S can be analysed...
L209: Ratios of elements
L210: I understand what you mean, but find it hard to follow. I suggest: However, stable isotope ratios in animal tissues may be influenced by complex physiological processes.
L213: ‘isotopic fractionation’ can be deleted for clarity
L213: Integration period -> isotopic ratios
L223: Suggest adding: When comparing...
L226: animal changed from -> How long the animal has spent as wild or captive (or vice versa)
L239: Analyses
L404: few -> The lack of suitable studies
L412: I know what you mean by wild vs captive but I think you could recap what you mean in a sentence so that there is no potential for confusion.
L413-417: This section could be removed as it is more of an introduction.
L420: I understand what you mean by various threat levels but it is not necessary here
L432: Stable isotope analyses were based...
L434: I'm not sure what "especially on a local scale means"? It could probably be removed.
L440: I'm not sure what is meant by this sentence - do you mean the accuracy of determination of whether n animal part is captive or wild?
L453: As with my comment for group 1: make sure that it is easy for a reader to tell what you mean by "wild and captive publications"
L461: In -> for
L462: Among -> between
L462: Better to put these percentages directly next to the categories they refer to.
L463: Not clear what this sentence means and what exactly is shown in fig 2 - please re-word.
L486: Maybe cite Phelps here too (Tensen is basically based on Phelps!)
L487-489: No need to cover this again
L489: SIA has been raised as a...
L568: Remove “high”
L569-571: I think this speaks to the use of SIA as a food safety tool for fish and not as a conservation tool. You could reword the sentence to make this point clearer.
L586: Remove "of each research" for simplicity of language.
L595: Animals'
L610-611: This sentence isn't clear and could be removed.
L612-613: What is meant by "some categories of wild and captive"? Fig 2 does not show categories it only shows wild or captive. This is fine as earlier you have chosen not to differentiate between categories of captivity and justified that - but don't at this stage introduce unnecessary complexity that is not shown in your figures!
L613-625: This section is mostly result not discussion
L628: Such a pattern
L631-632: This paragraph confusingly jumps from terrestrial animals to fish - I suggest discussing each taxonomic group separately. There is a lot of focus on fish generally and I don't see the overwhelming need - it would be nice to have more of a balanced discussion across taxonomic groups.
L648-650: This is a result. In general there are too many results in the discussion and too many papers are individually discussed. Focus on the results of your meta analysis and explain what that means for onwards use of stable isotope analysis.
L724: Throughout the discussion try to be specific - don't talk about "some studies": either include numbers in the results or don't discuss - it creates conclusion and detracts from the main points.
L740: Remove worldwide – there is no longer a spatial aspect to the results.
L746-750: Recap those points briefly here in the conclusion.
L752-755: How will this database help? I don't think that you should think of this as a database to be used, rather it is a meta-analysis that indicates that in the majority of cases, SIA has been able to differentiate between captive and wild.

Reviewer 2 ·

Basic reporting

The review is a good survey of the literature on the topic, but insufficient background information/context provided. The introduction could better introduce the topic and clearly define the study objectives.

Experimental design

Methods need more description in order to better understand what analyses were conducted, what the goal of each analysis was, and how the results help to address one of the stated objectives of the study. While the review covers most of the key papers in this area, there is a lack of critical analysis that is needed to move this beyond a summary of the state of the field.

Validity of the findings

The argument needs to be better developed in the introduction. The Conclusion does discuss some unresolved questions and suggestions for future directions, but could better analyze the results and present new information based on the review conducted. Specific comments provided below.

Additional comments

In this revision, the authors have made significant revisions to the original manuscript to shift the focus to wildlife farming, remove some of the analyses, and address the comments from the first round of reviews. While I acknowledge the effort put into this revision and agree that the paper is moving in the right direction, I unfortunately find that the arguments presented in this version of the manuscript remain somewhat under-developed and the review falls short of providing a clear critical analysis of the current state of the field. In addition, some of the analyses that raised concerns in the first round of reviews remain (particularly the analysis looking at average values across taxa, continent etc.) remain in this version without a clear justification in the text. As it currently stands, this version of the manuscript presents a summary of current studies on the topic rather than a critical review of the studies that helps to move the subject forward.

Introduction: This section has been revised to change the context of the paper to wildlife farming. As raised by Reviewer 1, there are many different types of captive breeding/rearing, of which wildlife farming is one – but this topic has not been explored sufficiently in the introduction. This is important as these different types of captivity might have impacts on the isotopic ratios measured – for example the stable isotope ratios in captive bred individuals could be quite different than those that were wild caught and captive reared.

I find that the main objective of the paper (and of the individual analyses conducted) is not clearly described. The main aim of the paper, as stated on the last line of the introduction, is to “evaluate the efficiency” of stable isotope analyses when trying to distinguish between wild and captive animals. However, it is not clear how exactly ‘efficiency’ is measured, nor how specifically it will be evaluated in this paper. The introduction could be restructured to provide background information specific to this objective.

Throughout the manuscript, the description of SIA (the theory and the methods) is not clear enough in many places to allow readers unfamiliar with this type of analysis to follow the main argument. To improve clarity, a more detailed (and focused) description of how SIA can be applied specifically to detect the origin of an individual (captive vs. wild) in the introduction would help. For example, fractionation is described briefly in the introduction, but there is no information on how fractionation impacts our ability to determine origin using stable isotopes. Similarly technical aspects of the stable isotope methods are mentioned (e.g. lipid extraction) without explaining why these would be done nor what the implications would be for the SIA results. I would recommend a thorough revision of the introduction to make sure that these key elements are clear.

The methods need more clear and detailed descriptions of overall study design (e.g. how does each test/analysis relate to the main objectives?). More detailed methods would help to understand what specific questions were being addressed and how each analysis was conducted (including what statistical test was used to address each question and what data was included in each test). For example, lines 196-199 discuss a series of analyses comparing means, SD and range of isotope values across taxa, continents, dietary guild etc, but it is not clear why this was done. How does this analysis help achieve the goal of testing for the efficiency of SIA to distinguish between captive and wild? What specifically was analyzed with regards to continent, taxon group and dietary category (categories need to be defined here)? From reading through the manuscript I assume this was primarily to provide an overview of where studies were conducted and on what taxa, but this needs to be explained here (and it would help to also include a sentence explaining the relevance of this – how does this summary help you to evaluate the efficiency of SIA? Later in the methods another series of analyses are referred to, but it is not clear what data were used with what statistical analysis and what specific questions was being teste with each analysis. The results for most analyses are reported qualitatively, not quantitatively (no statistics included in the results).

I provide some specific comments below which I hope will illustrate the general comments that I made above and help to guide a revision.

Lines 90-91 “Wildlife farming should meet a variety of biophysical, market and regulatory conditions to achieve conservation benefits.” - Not clear what these conditions are or how they relate to your current study. Your study focusses on a tool that can be used to distinguish between captive and wild animals and the introduction should focus on that aspect.
Line 110: I think you mean “life histories” not “life stories” here
Line 136-139: In these couple of sentences a series of factors that can influence the ratios of stable isotopes in an organism are listed. These are important factors that could impact our ability to use SIA to differentiate between captive and wild and should be treated more thoroughly. Would you be able to use SIA to differentiate between farming and ranching? Intensive and extensive farming? All of these terms should be clearly defined and the implications for stable isotope ratios explained.
Lines 140: References are need to support the assertion that most studies are based on the 'assumption' of more varied diets in wild vs. captive animals.
Line 182: All the terms here need to be defined and explained.
Lines 202: What was the reason for splitting the studies into these two groups (one where the analysis was explicitly to compare captive and wild, and one where data on captive and wild individuals was collected but not compared)? It seems like it would give you the most analytical power to analyze all of these studies together? How does splitting the studies this way help to achieve the main objective of the paper?
Lines 209-214: More detail on the types of statistical tests conducted is needed. What specific questions were being addressed (and how do these questions help to achieve the main objectives of the paper?) and which statistical test was applied to each dataset? It is not enough to say “tests such as t-test and ANOVA were used” – it should be explained what data was included in each statistical analysis and what specific question was being addressed by each test.
Line 249-250: In order to understand these results, more background is needed to explain why/how we might predict oxygen isotopes to vary in captive vs. wild populations.
Line 255: How does isotopic variation by feeding strategy relate to the objective of this paper? How do the results from these analyses help to evaluate the efficiency of SIA in this case?
Line 284: It is very hard to publish negative results, so this may be publication bias more than a confirmation that this technique works.
Line 307: Statistical results should be reported here. How were the 10 studies analyzed and what were the specific results of your analyses?
Discussion: The discussion currently lacks an in-depth synthesis of the results and reads more like a summary of the results of multiple papers without new insights provided by the analyses conducted from the study. It would be clearer if the discussion reflected back on the objectives/hypotheses outlined in the introduction to anlayze whether the obtained results support or refute those hypotheses. What overall conclusions can we draw from these studies, what key gaps remain, and what are the analytical challenges faced moving forward in this field?

Lines 366-370: Here it is reported that 80% of the studies could successfully distinguish between captive vs. wild animals using SIA. However, given that there may be some publication bias here (negative results being difficult to publish) I would be cautious in using this metric to support the argument that SIA is a good tool to answer this question. (I think SIA is a good tool, but there are stronger arguments to make).

---

## Round 0.3 · Minor Revisions

One reviewer was able to re-review your manuscript, and I am thankful for the expertise that was provided. At this point the only revisions boil down to language issues. Please see the edited manuscript (PDF) kindly provided by the reviewer. Also note that PeerJ provides language editing services, and it might be worth your time and effort to discuss that with the journal.

Thanks to everyone for their work on this manuscript, and particularly to the reviewers who have applied their volunteer time and expertise.

**Language Note:** The Academic Editor has identified that the English language must be improved. PeerJ can provide language editing services - please contact us at copyediting@peerj.com for pricing (be sure to provide your manuscript number and title). Alternatively, you should make your own arrangements to improve the language quality and provide details in your response letter. – PeerJ Staff

Reviewer 1 ·

Basic reporting

see comments

Experimental design

see comments

Validity of the findings

see comments

Additional comments

Firstly, my apologies for the delay in my review. This has been due to unavoidable personal circumstances. Should there be a lack of rigor by comparison to my previous reviews I am sorry and hope that there is sufficient guidance in this and my previous reviews to provide adequate direction to the authors.
I have reviewed the track-changes manuscript and the rebuttal letter. I think that this manuscript has evolved into a much clearer and well-defined form. The authors have compiled a database of existing stable isotope studies that deal with captive/wild differentiation and in this paper discuss general patterns and lessons that may be learned from these studies. The results are not unexpected or earth-shattering, but I believe that is the beauty of this piece. It now provides a solid foundational review of stable isotopes in captive/wild differentiation that caters to a broad range of expertise and interest groups. In my view a level of simplicity is to be admired and the authors have developed an excellent primer to the extent of the use of stable isotope analysis in addressing the differentiation of captive/wild in vertebrates. As a reviewer it is always tempting to ask for further analysis and depth of recommendations but in this case I won’t. This piece now delivers what it promises: a utilitarian summary of the published literature that will serve as a useful signpost to early and established researchers alike.
As per my previous review it could do with English language proof-checking and I have made suggestions (non-exhaustive) throughout. I again acknowledge the difficulty of writing in another language. The structure is clearer and the supplementary material more transparent. The conclusions are appropriate to the information presented and the discussion provides an appropriate overview rather than diving into specific detail.
I recommend (very) minor changes/publication.
My suggestions and comments are recorded as track changes in the document I upload.

Annotated reviews are not available for download in order to protect the identity of reviewers who chose to remain anonymous.

---

## Round 0.4 · accepted · Accept

This manuscript has gone through a good review process and it is nicely improved from the initial submission. The final minor revisions decision was based mainly on grammar, etc., and those issues have been satisfactorily resolved. As such, the manuscript is now ready for publication in PeerJ. Thanks to the reviewers and co-authors for their work in bringing this paper to this stage.